# The Genetic Structure of the Field Pea Landrace "Roveja di Civita di Cascia"

**Nicoletta Ferradini †, Renzo Torricelli †, Niccolò Terzaroli †, Emidio Albertini †**  **and Luigi Russi \*,†** 

Dipartimento di Scienze Agrarie, Alimentari e Ambientali, Università degli Studi di Perugia, 06100 Perugia, Italy; nicoletta.ferradini@unipg.it (N.F.); renzo.torricelli@collaboratori.unipg.it (R.T.); niccolo.terzaroli@studenti.unipg.it (N.T.); emidio.albertini@unipg.it (E.A.)

\* Correspondence: luigi.russi@unipg.it

† These authors contributed equally to this work.

**Abstract:** "Roveja di Civita di Cascia" is a landrace of *Pisum sativum* grown in marginal land habitats of the Apennines, Central Italy, and is one of the eleven herbaceous crop landraces listed in the Regional Register of local varieties. The objective of the present paper was to assess its genetic structure using 62 morphological traits and five microsatellites. As many as 55 traits showed significant differences with the control entries (*P. sativum* subsp. *sativum* var. *arvense* and var. *sativum*). We tested *P. sativum* Simple Sequence Repeats (SSRs) for their transferability to "Roveja", and found that only 12 out of 35 performed well. Of these, we demonstrated that five were sufficient to assess the genetic structure of this landrace, characterized by several private alleles, differentiating it from Paladio and Bluemoon, which were used as controls. Phenotypic and genotypic data evidenced a genetic structure based on a blend of several pure-bred lines. The sustainability of on-farm landrace conservation is discussed.

**Keywords:** field pea; genetic resources; genetic structure; landraces; local products; molecular markers; *Pisum sativum*; rural sustainability; SSR; Simple Sequence Repeats

## 1. Introduction

Pea (*Pisum sativum* L., family Fabaceae, genus *Pisum*, 2n = 14) is an annual herbaceous plant predominantly autogamous, native in the Middle East, between the Caucasus and Mesopotamia, with a secondary center of diversification in the Mediterranean Basin and in Ethiopia [1–3]. Several archeological evidences date its presence in the Middle East and Central Asia as early as 10,000 B.C., where, together with other legumes and cereals, it represented an important component of the diet of those civilizations [4–6]. Pea (field pea, *Pisum arvense*, and purple pea, *P. elatius*) and other Neolithic crops reached Europe through the migration or colonization of farmers and shepherds from the Near East [7]. Archeological sites and historical data report evidences of its cultivation since the Stone and Bronze Ages, where it seems to have represented, together with lentil, barley and spelt, the basis of animals and human nutrition [8].

The botanical classification of the genus *Pisum* has not yet been completely clarified. Originally, it was considered a genus composed by five species, later a monotypic genus, and more recently, a genus with two species [3,9–12]. According to the most recently accredited classification, the genus *Pisum* includes the wild species *P. fulvum* found in Jordan, Syria, Lebanon and Israel, and the cultivated species *P. abyssinicum* from Yemen and Ethiopia and *P. sativum* distributed throughout the world and consisting of both wild (*P. sativum* subsp. *elatius*) and cultivated subspecies (P. *sativum* L. subsp. *sativum*) [2].

Field pea (*Pisum sativum* subsp. *sativum* var. *arvense*, (L.) Poir.) is an annual legume crop widely used in the past for both human and animal consumption. Nowadays it is cultivated worldwide for

hay, pasture or silage production, either alone or in mixture with cereals. Seeds are rich in protein and minerals, and for these reasons, in northern Europe it is used as an alternative protein source in the feed industry. Similarly to many other grain legumes, the field pea is suitable in crop rotations due to the nitrogen fixation, especially in organic agricultural systems.

In Italy, until the first half of the last century, field pea was cultivated throughout the Apennine Ridge as a fresh and dry fodder for fattening rams and pigs, while occasionally being used also for human consumption. Nowadays a field pea landrace, locally known as "Roveja", "Roveglia" "Roveggia" or "Rubiglio", is residually cultivated in traditional and marginal farming systems and in the little rural context of the Sibylline area, a region between Umbria and Marche, in Central Italy. This legume, together with other local productions such as bean, lentil, emmer wheat, chickpea and grass pea, are at the base of marginal agricultural economies whose maintenance allows the development of typical products expressing and enhancing the culture of these territories [13–15]. The role and economic importance of the field pea in this region was clearly stated already in the 1545 statute of Montesanto di Sellano, imposing in every garden the cultivation of at least two legumes, including "Roveja" [16]. In Civita di Cascia, a little rural village in Valnerina, Central Italy, some farmers have successfully promoted its cultivation on a larger scale, and in 2004 it became first a Slow Food Presidium [17], and in 2015, one of the typical products of Regione Umbria, being included in the list of protected autochthonous genetic resources of agricultural interest [18].

Germplasm diversity is normally assessed by morphological descriptors that continue to be the only valid marker type accepted by the International Union for the Protection of New Varieties of Plants [19]. Germplasm characterization based on morphological characters facilitates the identification and selection of desirable traits, eventually transferring genes, such as resistance to biotic and abiotic stresses, into widely grown food legumes [20]. Since many morphological characters are quantitative and are influenced by environmental factors [21,22], the analysis of genetic diversity among pea local populations is realized on a combination of morphological traits and molecular markers [3,22–29].

The main objective of the present study was to assess the genetic structure of the field pea landrace "Roveja di Civita di Cascia", using morphological traits and molecular markers (SSRs). The knowledge of the genetic structure of this landrace, as highlighted through the estimation of allele frequencies within and among different lines, will also help the monitoring of changes in time that might occur as a result of selection pressures, including climate changes.

## 2. Materials and Methods

### 2.1. Plant Material

A preliminary study was conducted in 2004 in a farmer's field in Civita di Cascia, Perugia, Italy (42.671525 N, 13.120203 E), growing 120 spaced plants belonging to the original plant population of Roveja (*Pisum sativum* subsp. *sativum* var. *arvense*, (L.) Poir.). The progenies of plants with at least 35 seeds set (in order to have seed-rows of 10 plants, each replicated three times) were evaluated in 2011 (data not reported). Seeds of 43 lines randomly chosen from those obtained in 2011 (CC_lin) and of five control populations were used in the present study in 2013 and 2014. The controls are represented by the original Civita di Cascia landrace (CC_ori), two local accessions conserved in the Germplasm Bank of the University of Perugia: Castelluccio 3501 (CA) and Cermis 4767 (CE), the commercial, leafless Bluemoon variety (BM) of *P. sativum* subsp. *sativum* var. *arvense* and the commercial variety Paladio of *P. sativum* subsp. *sativum* var. *sativum* (PS). BM and PS are occasionally cultivated in nonmarginal areas of Central Italy.

Ten seeds per entry were sown in a polystyrene plateau, and after emergence, eight seedlings were transplanted in $20 \times 20$ cm pots (peat:sand, 70:30) and grown in a greenhouse, while 24 plants were used for the original seed lot of Civita di Cascia. Pots were arranged in a completely randomized design.

Twenty-four qualitative and thirty-eight quantitative traits were collected following the *P. sativum* International Union for the Protection of New Varieties of Plants (UPOV) guidelines [19], normally used for distinctness, uniformity and stability (Table A1). However, additional traits of agronomic

interest were also recorded, and for few of them, it was necessary to adapt the scores to some specific characteristics of var. *arvense*. Samples of leaves, stipules and seeds were scanned, and their measurements (size, area, perimeter) were obtained by ImageJ [30] and SmartGrain [31] software. Fine fragments of cotyledon tissues were extracted and placed onto a microscope slide, adding a droplet of water and gently squashing upon the coverslip. Simple starch grains are shaped like wheat seeds, while compound grains are star-shaped and appear to be made of a number of segments. Observations were carried out by a 20× optical microscope.

## 2.2. SSR Analysis

Five out of eight plants grown in pots were used for the genetic characterization of all entries; 21 plants were used for the original landrace Civita di Cascia, for a total of 253 individuals. Total genomic DNA was isolated from young leaves using the DNeasy® 96 Plant Kit (Qiagen, Hilden, Germany) according to the supplier's specifications.

A total of 35 genomic and EST SSRs specific for *P. sativum* [25,26,32–34] were tested. PCR reactions were performed in a total volume of 20 µL using: 1X reaction Buffer without $MgCl_2$, 1.5 mM $MgCl_2$, 200 µM dNTPs each, 0.4 µM each primer, 1 U Taq DNA polymerase (Sigma-Aldrich, St. Louis, Missouri, United States) and 20 ng genomic DNA. All amplifications were carried out with a GeneAmp PCR system 9700 (Applied Biosystems, Foster City, California, United States) programmed as follows: 94 °C for 4 min, followed by 30 cycles of 94 °C for 20 s, Ta (52–64 °C) for 30 s, 72 °C for 30 s and then 72 °C for 30 min. The annealing temperature was lowered by 2–5 °C according to the evolutionary distance among species, as suggested by Rossetto [35].

PCR products were separated on 2% agarose gel electrophoresis; in "Roveja" the SSR markers that did not amplify or were of unexpected size were discarded, while those showing polymorphic bands were selected and ligated into the pCR4-TOPO TA Vector (Invitrogen, Carlsbad, California, United States). Three positive clones for each SSR marker were selected for sequencing on an ABI Prism 3130 sequencer (Applied Biosystems, Foster City, California, United States) using a BigDye® Terminator V3.1 kit that employs a cycle sequencing protocol according to the manufacturer's specifications (Applied Biosystems, Foster City, California, United States). Vector sequences were removed, the unique sequences were edited and aligned with those of *P. sativum* using the sequence assembly program Vector NTI 9 Advance® (Invitrogen TM, Carlsbad, California, United States) and T-Coffee (Center for Genomic Regulation, Barcelona, Spain) [36], and subsequently screened for the presence of SSRs with the program Tandem Repeat Finder (Boston University, Massachusetts, United States) [37]. The SSR loci with sequences based on tandem repetitions were considered suitable for the genetic analysis (Table 1). Sequence editing was performed by Jalview 2.11 software [38].

**Table 1.** Characteristic of 15 *P. sativum* SSR markers sequenced in Roveja.

| Locus | Accession No. | Sequence Type | Position | Pea | | Roveja | |
|---|---|---|---|---|---|---|---|
| | | | | Motif | Alleles [†] | Motif | Alleles [‡] |
| PeaCPLHPPS [a] | L19651 | mRNA | 3'UTR | $(AT)_6$ | 4 | $(AT)_8$ * | 3 |
| AA430902 [a] | AA430942 | mRNA | CDS | $(AAT)_7$ | 3–4 | $(AAT)_7$ | |
| PSGAPa1 [a] | X15190 | mRNA | 3'UTR | $(AT)_{17}$ | 6–7 | $(AT)_{25}$ | 9 |
| PSP40SG [a] | X51594 | DNA | 5'UTR | $(AAT)_{36}$ | 5–8 | Not found | |
| PSMPAD134 [c] | - | - | - | TC/ATC | 4 | $(ATAG)_{32}$ | |
| AA321 [d] | Fj434429.1 | DNA | Transposon | $(TC)_{17} (AC)_{15}$ | 2 | $(TC)_{12}$ | 5 |
| PSMPB14 [d] | Fj434431.1 | DNA | Transposon | $(TC)_{22} (AC)_{25}$ | 3–9 | $(TC)_{24} (AC)_{21}$ * | |
| PSMPSAD186 [d] | - | - | - | TC/ATC | 3–8 | $(CT)_{17}$ | |
| PSMPSAD237 [c] | - | - | - | ATCT | 4–7 | $(AGAT)_{33}$ * $(AG)_7$ | 19 |
| Pea11 [e] | FG536955.1 | mRNA | - | $(ATGAAA)_6$ | 4 | Not found | |
| PSAD270 [c] | - | - | - | TC | 7–8 | $(CT)_{22} (ATCT)_4$ | 17 |
| PSMPSAA476 [b] | - | - | - | | 5–8 | $(TC)_{13}$ * $(CA)_{11}$ * | |
| PSMPSAA473 [b] | - | - | - | | 5 | $(GT)_{46}$ * $(CG)_8$ | |
| PSMPSAA278 [c] | - | - | - | GT | 7 | $(CA)_{12}$ | |
| PSRBCS3C [a] | X04334 | DNA | Intron 1 | $(AT)_6$ | 5 | Not found | |

[a] [32], [b] [26], [c] [33], [d] [25], [e] [34]; * imperfect; [†] number of alleles as reported in bibliography and [‡] in our study.

Primer sequence and allele range for validated loci were fed by Multiplex Manager [39] to determine the best sets of loci to include in a multiplex protocol. Multiplex Manager was used with the option of grouping all validated loci within the minimum number of PCRs, avoiding allele range overlap and primer interactions.

PCRs of the five selected loci were carried out with the Type-it Microsatellite PCR Kit (Qiagen, Hilden, Germany) containing 1X Type-it master mix with 0.2 μM of each fluorescent forward primer labeled with 6-FAM or ROX dyes (Sigma, St. Louis, Missouri, United States) and reverse unlabeled primer and 20 ng of template DNA and $H_2O$ to a final volume of 20 μL. All amplifications were performed in a GeneAmpPCRSystem 9700 (Applied Biosystems, Foster City, California, United States) consisting of a denaturing step at 95 °C for 5 min followed by 30 cycles at 95 °C for 30 s, Ta for 90 s and 72 °C for 30 s, and a final elongation step of 30 min at 60 °C.

PCR products were separated and analyzed on a 3130 XL DNA Analyzer (Applied Biosystems, Foster City, California, United States). The size of the amplified products was determined with an internal standard DNA (GeneScan 500 Liz, Thermo Fischer Scientific, Waltham, Massachusetts, United States) and the scorable peaks were assigned by GeneMapper software (Applied Biosystems, Foster City, California, United States).

## 2.3. Statistical Analyses

Qualitative traits were arranged in contingency tables and the difference among proportions was assessed by Pearson's chi-squared test. Quantitative traits were analyzed by univariate one-way Analysis of Variance (ANOVA). Principal Component Analysis (PCA) was performed using data of all traits and the number of principal components was determined using the minimum eigenvalue criterion recommended by Kaiser [40]. Univariate analyses were carried out by SAS software (Cary, NC, United States) [41], multivariate PCA by PAST software (University of Oslo, Norway) [42].

Prior to any statistical analysis of SSR data, common PCR artifacts leading to genotyping error were investigated. Presence of null alleles, large allele dropout and extreme stuttering was inferred by 1000 bootstraps and a 95% confidence interval (CI) using Micro-Checker 2.2.3 (Norwich Research Park Science, Norwich, United Kingdom) [43].

Observed (Ho) and expected (He) heterozygosity [44], F-statistics ($F_{IS}$ and $F_{ST}$) [45], analysis of molecular variance (AMOVA) [46,47] and Principal Coordinate Analysis (PCoA) [48] were performed using Genalex 6.5 software [49].

Bayesian model-based clustering implemented in STRUCTURE 2.3.3 (Pritchard Lab, Stanford University, United States ) [50,51] was performed to assess the genetic structure at the population level, as well as to detect genetic stocks contributing to this germplasm collection. Ancestry model with admixture and correlated allele frequency model was set to get the estimates of the posterior probability of data. STRUCTURE analysis was performed testing 20 independent runs with K from 1 to 21. The length of the burn-in period was set at 100,000, and the number of Markov Chain Monte Carlo (MCMC) repeats after burn-in were set at 500,000. The best K-value was determined through the 1K method [52] by using STRUCTURE HARVESTER 0.6.193 [53]. A membership coefficient qI ≥ 0.8 was used to assign the individuals to clusters. Individuals with membership coefficients qI < 0.8 were grouped together and considered "genetically admixed" [54–57].

## 3. Results

### 3.1. Phenotypic Characterization

The Chi-square test and the analysis of variance showed significant differences among entries in 55 out of 62 traits. Traits with no differences were: (i) Anthocyanin coloration in at least one part of the plant (leaves, stem or seeds), (ii) leaves and pod color (green for all entries), (iii) absence of pods parchment, (iv) presence of pod suture strings and (v) presence of stipule flecking. All details which referred to significant differences among entries are reported in Table A2. Compared to *P. sativum* var.

*sativum* Paladio (PS), all landrace lines of "Roveja" were characterized by longer stems, with more nodes, more branches and the presence of anthocyanin coloration.

Moreover, they showed more leaflets per leaf, but shorter and thinner, of different shape (the broadest part of the leaflet was located in the middle rather than at the base) and a more pronounced dentation. Stipules were also significantly smaller, in length and width, and hence, in area. Petioles were significantly shorter, although the variability among the "Roveja" lines was consistent, with some of them (CC_41, 26, 08, 11) not differing from Paladio. Except for lines 58 and 06, the landrace "Roveja" flowered significantly later and showed more flowers per node; the color of the wings was pink to reddish purple and the standards were cream rather than white. Other remarkable differences with Paladio were significantly shorter pods (almost half size) and with thinner walls; their shape at the distal part was essentially blunt while plants of Paladio showed a pointed end. Pod curvature was also slightly more pronounced in "Roveja". The mean number of ovule per pod was significantly lower (7.5 vs. 8.4, respectively), but the pod number per plant was higher (14.0 vs. 5.6, respectively), some of which were empty. In terms of seed starch, all accessions of "Roveja" showed simple grain starch compared with the compound grain starch of Paladio, and this was reflected on cotyledons wrinkling, absent or slight in the former, very marked in the latter (2.0 vs. 8.5, respectively). Other striking differences were on cotyledon color (orange in "Roveja", green and yellow in Paladio) and on testa marbling (very apparent in the former and almost absent in the latter). Interestingly, among the 43 accessions of "Roveja" there were significant differences for the hilum color, ranging from accessions whose color was the same as that of testa (CC_43, 37, 13) up to a very dark color (CC_38, 53, 42, 26). The different reproductive strategies between "Roveja" and PS, already shown as the number of ovules per pod and pods per plant, were confirmed also in terms of seed yield (4.3 vs. 6.6 g/plant), number of seeds per plants (51 vs. 29), 100-seed weight (10.0 vs. 24.3 g) and seed size (seed surface of 20.9 vs. 45.8 mm$^2$). Bluemoon, used as our control entry, although being a botanical variety *arvense*, for some traits (antochyanin stem color, stem length, color of the standard, pod wall, cotyledon color, testa marbling and 100-seed weight) was more similar to Paladio, for the others closer to the landrace "Roveja".

The first two principal components obtained from phenotypic traits (Figure 1) were able to explain almost the entire variance: 95.0 and 4.2%, respectively. The traits whose orthogonal dimensions were able to maximally separate the observations were the dimension of leaflets and of stipules (area, perimeter, length and width), positively correlated with the first component, and the number of pods and of seeds per plant positively correlated with the second component.

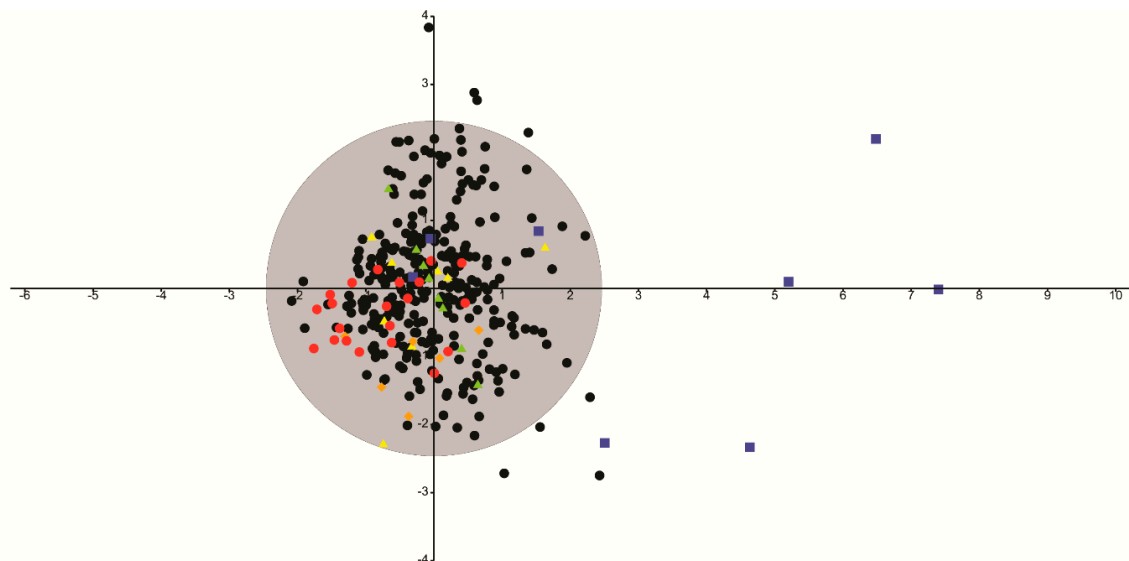

**Figure 1.** Principal Component Analysis (PCA) on phenotypic data. ● Civita di Cascia lines (CC_lin); ● Civita di Cascia original landrace (CC_ori); ▲ Castelluccio 3501 (CA); ◆ Cermis 4767 (CE); ▲ Bluemoon (BM); ■ Paladio (PS).

The gray region in Figure 1 circumscribes 95% of all individuals; eight individuals outside it were CC lines (three of which of CC_58) and five of them Paladio (PS_03, 08, 04, 02 and 06). The latter group of plants was scattered along the right side of PC1, characterized by a larger dimension of leaflets and petioles, thus confirming the differences found by univariate statistical analyses. From Figure 1 is also graphically evident that the total variability of "Roveja" lines was greater than that of CC_ori, the original seed lot, as well as of the CE and CA used as controls. This is important in conserving and managing landraces of inbred species because in our case 43 individuals were sufficient to preserve and retain enough diversity. This is confirmed also by examining for each trait the range of variation reported in Table A2.

### 3.2. Genotypic Characterization

A set of 35 pea genomic and EST SSRs were evaluated for their transferability from *P. sativum* to "Roveja". Genomic SSR markers are attractive, because they show a high level of polymorphism but, being unlinked to transcribed regions, do not have a defined gene function. On the contrary, EST–SSR are known for their high level of cross species transferability because of being located within transcribed regions of DNA characterized by a low mutation rate; for this reason they are more conserved but less polymorphic [58–60]. Although belonging to the same species, the pea SSR showed interesting differences, hereafter described. Out of 35 tested SSRs only 22 (62.8%) displayed clear and repeatable amplicons of expected or of approximate size, a surprising result considering the genetic proximity between the two taxa. Nonetheless, in Roveja only 15 of them were found polymorphic and therefore used for subsequent analysis. The 15 Roveja SSR loci were sequenced and aligned with the corresponding *P. sativum* sequences stored in the National Center for Biotechnology Information (NCBI) database, and the presence of SSR loci was confirmed in twelve of them (80%), revealing a high degree of conservation of SSR flanking regions in four of them: PeaCLHPPS, AA430902, PSMPB14 and PSGAPA1 (Figure S1).

Furthermore, the sequences revealed some differences in almost all analyzed loci, due to: (i) variations in length of the repetitive pattern and/or (ii) different repetitive motives (Table 1). For instance, comparing the Roveja and *P. sativum* motif at locus PSMPSAD237, the former contains a repeat of $(AGAT)_{33}$ $(AG)_7$, while the latter contains a repeat of ATCT; locus AA321 shows a $(TC)_{12}$ motif in the former vs. a $(TC)_{17}$ $(AC)_{15}$ motif in the latter; and locus PSAD270 contain a $(CT)_{22}$ $(ATCT)_4$ motif vs. a TC motif [25]. Finally, in some loci the same type of repeat resulted conserved across the two taxa. Loci PeaCLHPPS and AA430902 exhibited the same length and repetitive motives in Roveja and in pea, whereas loci PSMPB14 and PSGAPA1 showed differences in the length variations of the microsatellite repeats. On the other hand, it is well known that SSR polymorphism is the result of differences in the number of repeats of the motif caused by polymerase strand-slippage in DNA replication, unequal crossing-over or by recombination errors [61,62]. Therefore, different individuals, and even more, different species, exhibit variations as differences in repeat numbers.

Despite the autogamous nature of the reproductive system and of only five SSR markers used, we found a great variability in terms of the number of alleles among the analyzed loci, particularly for PSMPSAD237 and PSAD270, (with 19 and 17 alleles, respectively, Table 1). This result could be explained by the intrinsic characteristics of SSR markers, in general characterized by high mutation rates [63,64], positively correlated with the length of the repetitive motif rather than to the motif itself. Therefore, SSRs with a large number of repeats, as in our case, are more inclined to mutate due to the increased probability of slippage [62,65–67].

The five nuclear SSRs produced scorable amplicons with a total of 53 alleles showing an average polymorphism of 70%. The average number of alleles per locus was as low as 3.33 (ranging from 1 to 12) and the number of effective alleles per locus was even lower (Ne = 2.22) (Table 2).

**Table 2.** Polymorphic loci, number of alleles (Na) and of effective alleles (Ne), observed (Ho) and expected (He) heterozygosity and inbreeding coefficients (F) of the entries as assessed by five SSR markers.

| Entries | Polymorphic Loci (%) | Na | Ne | Ho | He | F |
|---|---|---|---|---|---|---|
| CC_lin (all) | 100 | 6.8 | 3.17 | 0.08 | 0.52 | 0.84 |
| CC_ori | 100 | 6.0 | 3.68 | 0.08 | 0.54 | 0.85 |
| CA | 60 | 2.4 | 2.33 | 0.08 | 0.38 | 0.79 |
| CE | 80 | 2.2 | 1.75 | 0.05 | 0.37 | 0.86 |
| BM | 40 | 1.2 | 1.09 | 0.20 | 0.16 | 0.00 |
| PS | 40 | 1.4 | 1.26 | 0.05 | 0.14 | 0.65 |
| Mean | **70** | **3.3** | **2.22** | **0.09** | **0.35** | **0.69** |
| SE | 11.3 | 0.59 | 0.337 | 0.035 | 0.055 | 0.087 |

Several lines (CC_05, 08, 18, 21, 22, 31, 33, 35, 36, 43, 55, 58, 59, 62, 64 and 67) were monomorphic at all five loci considered. Considering only the polymorphic loci, the observed heterozygosity (Ho) was nil in 7 cases out of 30, and 17 entries were not in Hardy–Weinberg equilibrium due to the lower proportion of heterozygote compared to the expected. Therefore, despite the fact that the color of the flower is able to attract pollinators, "Roveja" showed high rates of selfing (mean F values 0.69 and $F_{IS}$ = 0.68).

Nevertheless, beside the low gene flow due to the reproductive system, the high level of gene fixation is also attributable to genetic drift, with an $F_{ST}$ value of 0.50, indicating very strong differentiation among entries (Table 3).

**Table 3.** F-statistics obtained by SSR markers.

| Locus | FIS | FST | FIT |
|---|---|---|---|
| PSGAPa1 | −0.022 | 0.456 | 0.444 |
| PSMPSAD237 | 0.965 | 0.414 | 0.979 |
| PSAD270 | 0.854 | 0.332 | 0.902 |
| PeaCPLHPPS | 0.675 | 0.433 | 0.816 |
| AA321 | 0.934 | 0.856 | 0.991 |
| Mean | 0.681 | 0.498 | 0.826 |
| SE | 0.183 | 0.092 | 0.101 |

By examining the result of PCoA (Figure 2), it is also evident that the differentiation is not among entries of "Roveja" (CC_lin, CC_ori, CA and CE), but between these and PS and BM. The two coordinates were able to explain as much as 98% of total variation, with Coordinate 1 able to differentiate the "Rovejas" from BM and Coordinate 2 from PS.

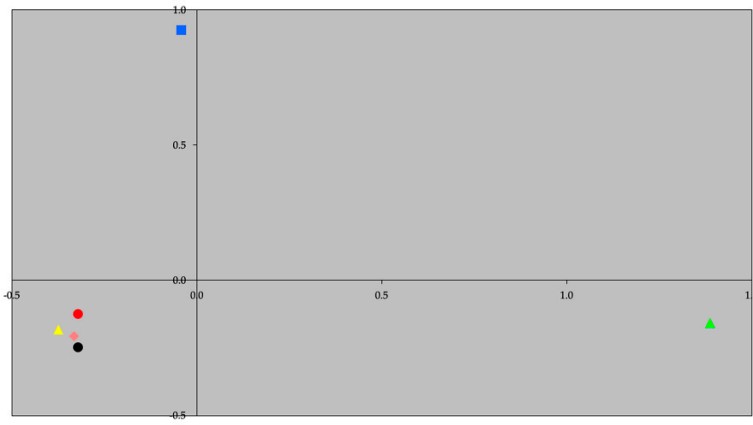

**Figure 2.** Principal Coordinate Analysis on genotypic data. ● Civita di Cascia lines (CC_lin); ● Civita di Cascia original landrace (CC_ori); ▲ Castelluccio 3501 (CA); ◆ Cermis 4767 (CE); ▲ Bluemoon (BM); ■ Paladio (PS).

The five SSRs were also used to determine the genetic structure among entries. The average log-likelihood values for Ks from 1 to 20 and the distribution of ΔK values [52] indicated two peaks, corresponding to K = 2 and K = 14. The hierarchical genetic structure was investigated at K = 14 and a threshold value qI ≥ 0.80 was used to assign individuals to the clusters (Figure 3).

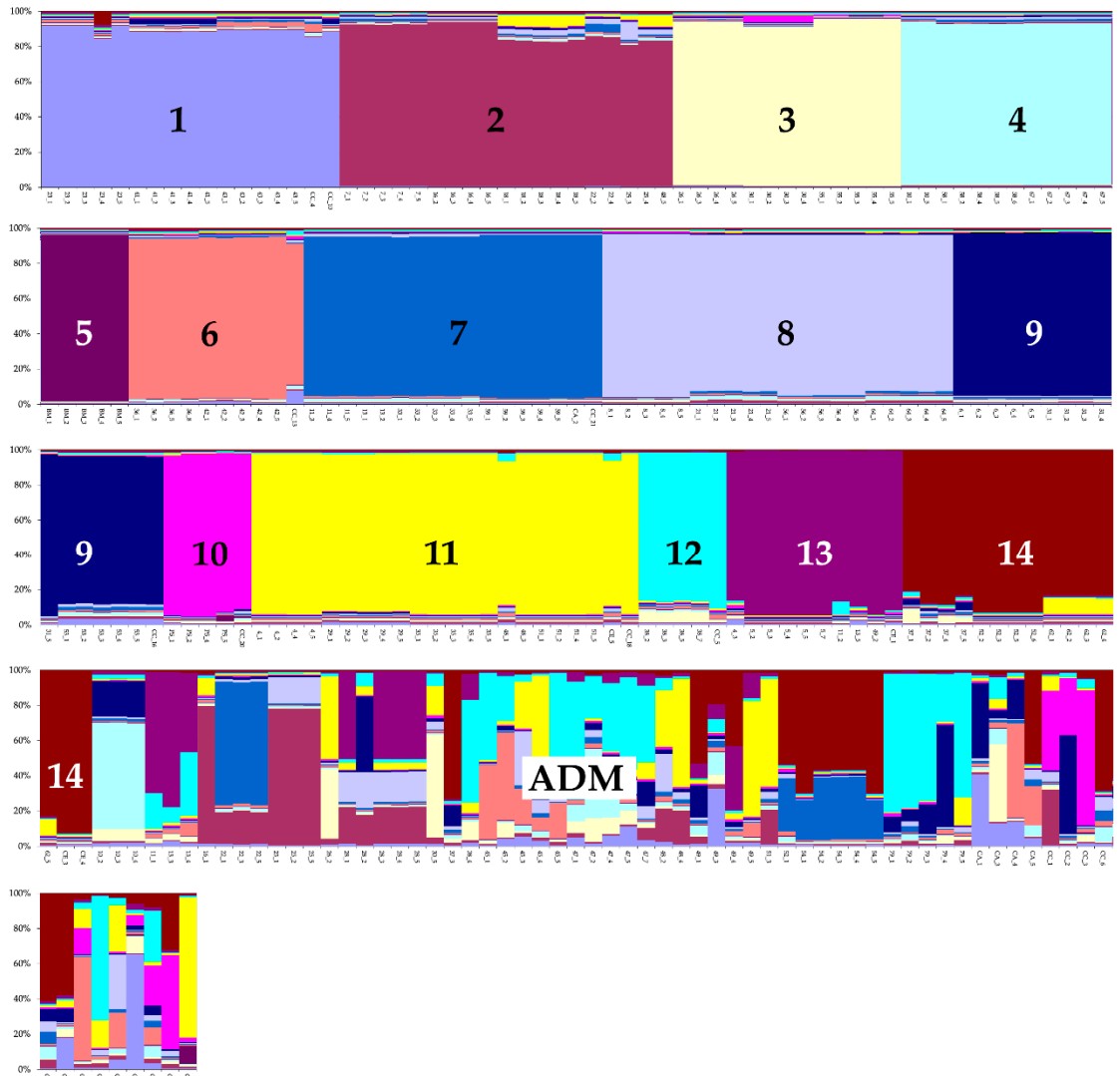

**Figure 3.** Graphical representation of STRUCTURE results. The numbers within differently colored groups indicate the cluster number using a membership coefficient qI ≥ 0.8. Individuals not included in any cluster are considered genetically admixed (ADM).

Out of 253 individuals, 67 (26%) were not classified in any of the 14 clusters and classified in the genetically admixed group (ADM). The group ADM included all individuals from several lines of "Roveja", from CC and CA. With the exception of Cluster 12, all the others included at least one complete set of individuals belonging to the same line/entry (Figure 3).

Cluster 1 included all individuals of three lines (CC_lin 23, 41 and 43); Cluster 2 included line 07 and 18; Cluster 3 line 55; Cluster 4 line 58 and 67; Cluster 5 included only individuals of BM; Cluster 6 was based only on line 42; Cluster 7 included lines 33 and 59; Cluster 8 lines 08, 21, 56 and 64; Cluster 9 lines 06, 31 and 53; Cluster 10 included all plants of PS; Cluster 11 all plants of line 29 and 35; Cluster 13 line 05 and Cluster 14 line 62.

Individuals of CC_ori were allocated into several clusters (1, 6, 7, 9, 10, 11, 12 and ADM), similarly to those of CA (cluster 7 and ADM) and CE (cluster 11, 13 and 14). In summary, combining the

results of allele frequencies, F-statistics and STRUCTURE, it is evident that the genetic structure of the landrace "Roveja di Civita di Cascia" is based on numerous lines, mostly homozygote at several loci, significantly different from BM and PS, but intermixed with the original population CC_ori and with CA and CE.

The AMOVA showed that the within individual variation was 13%, among entries was 19%, while most of it (68%) was among individuals within entries. In fact, the differences among individuals within entries are likely to be due to homozygous, fixed alleles at different loci.

Out of 53, as many as 26 alleles were private, i.e., unique to the various entries (Table 4), confirming the low gene flow among populations, and at the same time, explaining the goodness of differentiations among entries that STRUCTURE was able to evidence.

In summary, the landrace "Roveja di Civita di Cascia" is composed by several pure bred lines and the environmental conditions, the farming system and the mating system are likely to be key factors in determining its genetic structure.

**Table 4.** Private alleles found in pea entries for each locus. The values are base pairs.

| Pop | No. of Plants | PSGAPa1 | PSMPSAD237 | PSAD270 | AA321 |
|-----|-----|-----|-----|-----|-----|
| CC_lin | 35 | 192, 194 | 253, 257, 263, 306 | 266, 269, 297 | 387 |
| CC_ori | 9 | 184 | 234, 246, 280, 314, 366 | 228, 255 | 379 |
| BM | 5 | 178, 180 | 350 | 286, 288 | - |
| PS | 4 | - | - | 251, 279 | - |

## 4. Discussion

Although our landrace Roveja and pea belong to the same species, the results of the present study showed differences concerning phenotypic, genetic and genotypic aspects. In the first case the differences with the control variety Paladio emerged from both univariate and multivariate analyses. In the second case there were differences in the motif of the SSR repetitions, and in the third case, differences in the number of repetitions. These results confirm the on-going debate on the botany of the genus *Pisum*.

The case of "Roveja di Civita di Cascia" is emblematic, because a number of crops grown in Central Italy have the same history and a similar social context. In the 1960s, many farmers from hilly and mountain areas of South and Central Italy moved to large cities and to industrial areas in the North of the country. This caused a consistent abandonment of marginal lands and a parallel loss of agricultural genetic diversity. Nevertheless, the landraces of several crops remained part of the farming systems due to rural traditions linked to food and feed [68–70]. Fortunately, the risk of the genetic erosion of local germplasm was opportunely recognized earlier than other countries [14], and in the 1990s, four Regions of Central Italy (Tuscany, Umbria, Latium and Marche) promulgated laws aimed at collecting and conserving ex-situ and in-situ genetic resources. A detailed inventory of available accessions was published by Negri in 2003 [71], and recent studies confirmed the richness of the agricultural biodiversity of this area due to peculiar soil, climatic and social conditions [15,72–76]. These laws are concerned with the use of genetic resources in relation to the rural development. In particular, the relationship between genetic resources, territory, typical products and local traditions are a focal point in establishing political measures aiming at preserving crop landraces.

Sustainability in agriculture can be declined in several ways: (i) type and amounts of external inputs (fertilizers, pesticides, water), (ii) suitable exploitation of modern technologies (tools improving water use efficiency, equipment for a precision agriculture), (iii) economic return for a single crop, for a given farm, or for a whole region; (iv) environmentally sound practices, etc. Dealing with living organisms, and particularly under marginal agricultural conditions, sustainability must be entrusted to the genetic variability of plants, animals and microorganisms, paying particular attention to its extent, nature and evolution.

Within this framework the size of the variability of "Roveja di Civita di Cascia" was shown to be adequate, and it is likely to remain so if it continues to be grown according to the Production Regulations [77] approved by the farmer's association. Its genetic variability was evident, and all the same, enclosed within boundaries sufficient to reveal its uniqueness: in four out of five SSR markers we found as many as 10 private alleles. This result is of importance because it grants traceability in case of frauds, and this can be done at low costs. Moreover, the alleles frequencies can be monitored in time, assessing their changes caused by natural evolutive factors (gene flow, mutations, genetic drift, selection) and anthropic interventions. The former are expected to occur normally, but also as a result of the rapid changes of climatic factors. The latter are likely to occur in the event of breeding programs aiming at improving the productivity.

In such circumstances it is recommended to carry out a mass selection, discarding few plants with evident symptoms of diseases and/or with few pods rather than selecting few superior genotypes. In all cases, care must be taken in monitoring the changes in the overall variability, including distinctive traits.

The case examined in the present paper is one example of on-farm conservation of local landraces. The role played by farmers in conserving such valuable genetic resources is difficult to quantify economically. In such cases, public institutions can provide incentives as subsidies to the farmers to safeguard this kind of crops. This form of assistance can guarantee the safeguard in the short period, but it is not sustainable forever, as it is dependent upon public funds. The case of "Roveja di Civita di Cascia" is an example of an efficient germplasm conservation, as nowadays the crop provides enough income to the rural community.

The average production of Roveja is about 0.7–1.0 t ha$^{-1}$ compared to 1.5–2 t ha$^{-1}$ of commercial varieties of pea used as feed (e.g., Bluemoon), and the average price of the former is around 15–20 € kg$^{-1}$ compared to about 20 cents of the latter. Even considering the higher production costs of Roveja (packaging, labeling, marketing, etc.), the economic return for the rural community is remarkable and gives sustainability to the whole system. The promotion efforts put in place were appreciated by the market, so that in 2004 Roveja was awarded the Slow Food Presidium.

Rather than relying on subsidies, the model of "Roveja di Civita di Cascia" should be followed by many other local crops: The higher market prices compared to common products justify not only the safeguarding of the environment and of genetic resources but also local traditions, strenghtening the links of the rural community to its land.

**Supplementary Materials:** The following are available online at http://www.mdpi.com/2071-1050/11/22/6493/s1, Figure S1: SSR and flanking region sequences alignment of Roveja and Pisum.

**Author Contributions:** The Authors equally contributed to the whole work, from conceptualization to methodology, and from statistical analysis to writing.

**Funding:** This research received no external funding.

**Acknowledgments:** Silvana Crespi and Adelino De Carolis Farm is acknowledged for providing the original population of Roveja and the lands used in the experiments of 2004.

**Conflicts of Interest:** The authors declare no conflict of interest.

## Appendix A

**Table A1.** List of traits recorded on several accessions of the field pea landrace Roveja, and several controls, following the International Union for the Protection of New Varieties of Plants (UPOV) recommendations for *Pisum sativum* with slight changes to adapt to var. *arvense*.

| UPOV | Trait | Code |
| --- | --- | --- |
| 1 | Plant: anthocyanin coloration (1 = absent; 9 = present) | PlAntCol |
| 2 | Stem: anthocyanin coloration of axil (1 = absent; 2 = single ring; 3 = double ring) | StAnCol |
|  | Stem: intensity of color of axil (3 = light; 5 = medium; 7 = dark) | StAnInt |
| 3 | Stem: fasciation (1 = absent; 9 = present) | StFasc |
| 4 | Stem: length (cm) | StLenght |

**Table A1.** *Cont.*

| UPOV | Trait | Code |
|------|-------|------|
| 5 | Stem: number of nodes up to and including first fertile node (n) | STNnode |
|  | Stem: number of apex on the main stem (n) | StNApex |
| 6 | Foliage: color (1 = yellow green; 2 = green; 3 = blue green) | FolColor |
| 7 | Foliage: intensity of color (3 = light; 5 = medium; 7 = dark) | FolIntCo |
| 8 | Leaf: leaflets (1 = absent; 9 = present) | LeaLflt |
| 9 | Leaf: number of leaflets (n) | LeNLeafl |
| 11 | Leaflet: length (mm) | LftLeng |
| 12 | Leaflet: width (mm) | LftWid |
| 10 | Leaflet: size (area, mm$^2$) | LflArea |
|  | Leaflet: perimeter (mm) | LflPerim |
| 13 | Leaflet: position of broadest part (1 = middle; 2 = towards base; 3 = at the base) | LflBroad |
| 14 | Leaflet: dentation (1 = absent; 9 = very strong) | LflDent |
| 15 | Stipule: length (mm) | StpLeng |
| 16 | Stipule: width (mm) | StpWidt |
| 18 | Stipule: length from axil to tip (mm) | StpLenT |
| 17 | Stipule: size (area, mm$^2$) | StpArea |
|  | Stipule: perimeter (mm) | StpPerim |
| 20 | Stipule: flecking (1 = absent; 9 = present) | StpFleck |
| 21 | Stipule: density of flecking (1 = very sparse; 5 = medium; 9 = very dense) | StpFlDen |
| 22 | Petiole: length from axil to first leaflet or tendril (cm) | PtlLenl |
| 23 | Petiole: length from axil to last tendril (cm) (only varieties with leaflets absent) | PtlLent |
|  | Time of flowering: days from sowing to appearance of the first flower bud | Flowr1 |
| 24 | Time of flowering: days from sowing to appearance of the first five flowers | FlwrFl |
| 25 | Plant: maximum number of flowers per node (n) | PlFlwNd |
| 26 | Flower: color of wing (1 = white with pink blush; 3 = pink; 5 = reddish purple) | FlwWnC |
| 27 | Flower: color of standard (1 = white; 2 = whitish cream; 3 = cream) | FlwStC |
| 32 | Flower: shape of apex of upper sepal (1 = acuminate; 3 = acute; 5 = rounded) | FlwSeSh |
|  | Peduncle: spur (1 = absent; 9 = present) | PedSpur |
| 33 | Peduncle: length of spur (mm) | PedSpLe |
| 34 | Peduncle: length from stem to first pod (mm) | PedLeng1 |
| 35 | Peduncle: length between first and second pods (mm) | PedLeng2 |
| 37 | Pod: length (mm) | PodLeng |
| 38 | Pod: width (mm) | PodWidt |
| 39 | Pod: parchment (1 = absent or partial; 9 = present) | PodParch |
| 40 | Pod: thickened wall (1 = absent; 9 = present) | PodWall |
| 41 | Pod: shape of distal part (1 = pointed; 2 = blunt) | PodShap |
| 42 | Pod: curvature (1 = absent; 5 = medium; 9 = very strong) | PodCurv |
| 43 | Pod: color (1 = yellow; 2 = green; 3 = blue green; 4 = purple) | PodColor |
| 45 | Pod: suture strings (1 = absent; 9 = present) | PodSutur |
| 46 | Pod: number of ovules (n) | PodOvul |
|  | Pod: number per plant | PodNum |
|  | Pod: number of empty pods | PodEmpt |
| 47 | Immature seed: intensity of green color (1 = light 3 = medium; 5 = dark) | SeedIntC |
| 48 | Seed: shape | SeedShp |
| 49 | Seed: type of starch grains (1 = simple; 2 = compound) | SeedStrc |
| 50 | Seed: wrinkling of cotyledon (1 = absent; 3 = min; 9 = very strong | SeedWrk |
| 52 | Seed: color of cotyledon (1 = green; 2 = yellow; 3 = orange) | SeedCoC |
| 53 | Seed: marbling of testa (1 = absent; 3 = medium; 5 = strong) | SeedMar |
| 55 | Seed: hilum color (1 = same as testa; 5 = darker than testa; 9 = very dark) | SeedHilC |
|  | Seed: number of seeds per plant(n) | SeedNPla |
|  | Seed: yield per plant(g) | SeedYld |
| 57 | Seed: weight of 100 seeds (g) | HSW |
|  | Seed: length (mm) | SeedLen |
|  | Seed: width (mm) | SeedWid |
|  | Seed: surface (mm$^2$) | SeedArea |
|  | Seed: perimeter (mm) | SeedPeri |
|  | Seed: ratio length-width | SeedLW |

**Table A2.** Mean values per accession relative to all morpho-physiological traits examined. Significant differences among entries for quantitative traits (Analysis of Variance (ANOVA), F-test) were separated by the least significant difference (LSD), while for qualitative traits by Pearson's Chi-squared test (*P* = probability level); n/r = data not recorded).

| Accession | StAnCol | StFasc | StLenght | STNnode | StNApex | FolIntCo | LeaLflt | LeNLeafl | LfltLeng | LfltWid | LflArea | LflPerim |
|-----------|---------|--------|----------|---------|---------|----------|---------|----------|----------|---------|---------|----------|
| CC 04 | 9 | 1 | 149.4 | 28.0 | 1.4 | 5.0 | 9 | 6.0 | 32.9 | 14.1 | 349.2 | 82.2 |
| CC 05 | 9 | 1 | 164.9 | 30.3 | 3.3 | 5.0 | 9 | 5.5 | 36.5 | 18.3 | 517.9 | 93.0 |
| CC 06 | 9 | 1 | 153.8 | 28.3 | 3.0 | 5.0 | 9 | 5.5 | 36.4 | 18.7 | 518.4 | 93.4 |
| CC 07 | 9 | 1 | 160.7 | 29.3 | 4.0 | 5.0 | 9 | 6.0 | 42.1 | 17.9 | 588.0 | 105.2 |
| CC 08 | 9 | 1 | 152.5 | 28.7 | 3.0 | 5.0 | 9 | 4.8 | 41.8 | 20.8 | 678.4 | 108.0 |
| CC 10 | 9 | 1 | 162.9 | 31.0 | 4.1 | 5.0 | 9 | 7.1 | 40.9 | 19.3 | 592.3 | 102.5 |
| CC 11 | 9 | 1 | 174.2 | 30.8 | 4.8 | 5.0 | 9 | 5.8 | 39.7 | 20.2 | 607.1 | 101.9 |
| CC 13 | 9 | 1 | 145.4 | 29.2 | 3.8 | 5.0 | 9 | 6.0 | 35.1 | 18.8 | 522.1 | 93.4 |
| CC 16 | 9 | 1 | 146.5 | 29.8 | 4.4 | 5.0 | 9 | 6.3 | 39.7 | 18.0 | 543.5 | 99.0 |
| CC 18 | 9 | 1 | 147.1 | 29.0 | 2.8 | 5.0 | 9 | 6.0 | 40.6 | 20.3 | 597.3 | 102.3 |
| CC 21 | 9 | 1 | 150.3 | 26.3 | 2.5 | 5.0 | 9 | 6.0 | 36.3 | 16.8 | 474.4 | 91.5 |
| CC 22 | 9 | 1 | 132.8 | 29.2 | 2.7 | 5.0 | 9 | 6.0 | 45.5 | 21.9 | 739.1 | 114.7 |
| CC 23 | 9 | 1 | 157.8 | 32.6 | 3.3 | 5.0 | 9 | 5.5 | 36.2 | 18.7 | 488.3 | 91.6 |
| CC 25 | 9 | 1 | 129.9 | 29.0 | 2.5 | 5.0 | 9 | 6.0 | 38.6 | 19.8 | 564.7 | 98.4 |
| CC 26 | 9 | 1 | 156.7 | 29.4 | 3.3 | 5.0 | 9 | 5.9 | 39.4 | 21.6 | 620.9 | 102.0 |
| CC 28 | 9 | 1 | 136.6 | 30.3 | 3.6 | 5.0 | 9 | 6.0 | 42.3 | 18.7 | 576.4 | 102.3 |
| CC 29 | 9 | 1 | 149.3 | 28.0 | 3.9 | 5.0 | 9 | 6.1 | 38.7 | 18.0 | 504.7 | 95.6 |
| CC 30 | 9 | 1 | 144.5 | 29.8 | 2.5 | 5.0 | 9 | 5.8 | 37.3 | 21.0 | 589.5 | 98.9 |
| CC 31 | 9 | 1 | 170.5 | 28.6 | 4.1 | 5.0 | 9 | 6.3 | 36.5 | 18.8 | 520.0 | 95.6 |
| CC 33 | 9 | 1 | 88.5 | 23.0 | 1.8 | 5.0 | 9 | 4.5 | 35.4 | 20.4 | 542.4 | 93.3 |
| CC 35 | 9 | 1 | 151.4 | 26.0 | 4.1 | 5.0 | 9 | 6.0 | 30.6 | 14.9 | 378.7 | 78.1 |
| CC 36 | 9 | 1 | 153.1 | 28.3 | 2.5 | 5.0 | 9 | 6.0 | 38.8 | 19.8 | 558.5 | 98.0 |
| CC 37 | 9 | 1 | 148.1 | 28.1 | 2.5 | 5.0 | 9 | 5.9 | 41.3 | 20.7 | 615.2 | 102.3 |
| CC 38 | 9 | 1 | 142.0 | 31.7 | 3.3 | 5.0 | 9 | 6.0 | 38.9 | 21.5 | 642.2 | 101.6 |
| CC 41 | 9 | 1 | 181.5 | 31.1 | 2.5 | 5.0 | 9 | 6.0 | 35.3 | 16.6 | 437.6 | 89.1 |
| CC 42 | 9 | 1 | 159.9 | 31.4 | 2.1 | 5.0 | 9 | 6.0 | 40.1 | 21.3 | 601.1 | 100.9 |
| CC 43 | 9 | 1 | 156.8 | 29.8 | 4.8 | 5.0 | 9 | 5.9 | 41.7 | 19.4 | 608.2 | 103.9 |
| CC 45 | 9 | 1 | 158.8 | 31.1 | 4.9 | 5.0 | 9 | 6.0 | 36.9 | 16.3 | 502.1 | 94.5 |
| CC 47 | 9 | 1 | 144.4 | 30.9 | 3.5 | 5.0 | 9 | 6.0 | 36.6 | 18.3 | 517.6 | 94.1 |
| CC 48 | 9 | 1 | 127.5 | 28.1 | 3.8 | 5.0 | 9 | 6.3 | 37.6 | 18.2 | 509.9 | 94.6 |
| CC 49 | 9 | 1 | 153.3 | 28.2 | 6.5 | 5.0 | 9 | 7.2 | 33.9 | 17.3 | 455.1 | 88.3 |
| CC 51 | 9 | 1 | 144.3 | 26.1 | 3.6 | 5.0 | 9 | 6.4 | 38.8 | 19.1 | 534.8 | 96.4 |

Table A2. *Cont.*

| Accession | StAnCol | StFasc | StLenght | STNnode | StNApex | FolIntCo | LeaLflt | LeNLeafl | LfltLeng | LfltWid | LflArea | LflPerim |
|---|---|---|---|---|---|---|---|---|---|---|---|---|
| CC 52 | 9 | 1 | 153.9 | 28.5 | 3.8 | 5.0 | 9 | 5.5 | 43.8 | 23.4 | 737.7 | 110.5 |
| CC 53 | 9 | 1 | 141.3 | 27.9 | 3.6 | 5.0 | 9 | 6.3 | 33.4 | 14.6 | 378.2 | 83.7 |
| CC 54 | 9 | 1 | 142.3 | 28.8 | 4.3 | 5.0 | 9 | 4.6 | 35.3 | 19.2 | 556.5 | 96.7 |
| CC 55 | 9 | 9 | 129.9 | 31.5 | 4.8 | 5.0 | 9 | 5.9 | 43.5 | 21.7 | 729.5 | 111.7 |
| CC 56 | 9 | 1 | 156.9 | 28.9 | 3.0 | 5.0 | 9 | 6.0 | 33.5 | 16.6 | 404.1 | 84.0 |
| CC 58 | 9 | 1 | 124.0 | 23.5 | 1.3 | 5.0 | 9 | 5.8 | 40.4 | 24.0 | 730.6 | 108.0 |
| CC 59 | 9 | 1 | 139.9 | 27.7 | 3.7 | 5.0 | 9 | 6.0 | 32.3 | 16.6 | 381.7 | 89.1 |
| CC 62 | 9 | 1 | 152.1 | 29.8 | 6.0 | 5.0 | 9 | 6.1 | 37.4 | 20.6 | 530.7 | 94.4 |
| CC 64 | 9 | 1 | 137.2 | 29.2 | 6.2 | 5.0 | 9 | 6.0 | 35.2 | 17.0 | 529.0 | 95.3 |
| CC 67 | 9 | 1 | 151.4 | 28.1 | 4.8 | 5.0 | 9 | 6.0 | 39.1 | 19.4 | 705.7 | 108.4 |
| CC 79 | 9 | 1 | 140.3 | 29.5 | 4.1 | 5.0 | 9 | 6.0 | 36.6 | 18.4 | 549.8 | 97.7 |
| CC_ori | 9 | 1 | 133.4 | 26.7 | 3.0 | 5.0 | 9 | 6.2 | 32.4 | 15.6 | 392.1 | 82.3 |
| CA | 9 | 1 | 138.0 | 29.5 | 4.9 | 5.0 | 9 | 6.3 | 34.5 | 18.6 | 501.7 | 90.8 |
| CE | 9 | 1 | 134.9 | 28.3 | 3.4 | 5.0 | 9 | 6.0 | 32.4 | 16.8 | 414.0 | 83.2 |
| BM | 1 | 1 | 86.4 | 24.6 | 1.4 | 5.0 | 1 | n/r | n/r | n/r | n/r | n/r |
| PS | 1 | 1 | 79.4 | 16.8 | 1.1 | 6.3 | 9 | 5.1 | 47.0 | 30.1 | 1234.8 | 137.9 |
| LSD | | | 18.63 | 2.84 | 1.61 | | | 0.50 | 5.78 | 3.78 | 185.7 | 15.35 |
| P< | 0.001 | 0.001 | | | | 0.001 | 0.001 | | | | | |

| Accession | LflBroad | LflDent | StpLeng | StpWidt | StpLenT | StpArea | StpPerim | StpFlDen | PtlLenl | Flowr1 | FlwrFl |
|---|---|---|---|---|---|---|---|---|---|---|---|
| CC 04 | 2.0 | 6.2 | 50.1 | 23.3 | 36.4 | 848.8 | 144.2 | 3.0 | 5.3 | 115.4 | 122.4 |
| CC 05 | 1.1 | 5.5 | 53.9 | 26.2 | 37.7 | 1018.4 | 150.9 | 4.8 | 5.2 | 116.3 | 124.0 |
| CC 06 | 1.3 | 4.3 | 60.4 | 30.0 | 44.4 | 1368.2 | 172.4 | 3.5 | 4.4 | 101.1 | 111.9 |
| CC 07 | 1.1 | 5.6 | 63.2 | 29.2 | 42.8 | 1276.7 | 171.4 | 3.0 | 5.2 | 113.7 | 120.1 |
| CC 08 | 1.3 | 6.3 | 65.0 | 31.4 | 45.3 | 1467.6 | 183.4 | 2.7 | 5.7 | 116.7 | 124.0 |
| CC 10 | 1.5 | 2.0 | 63.1 | 32.6 | 46.3 | 1479.0 | 169.2 | 4.5 | 4.9 | 119.8 | 127.6 |
| CC 11 | 1.2 | 7.0 | 58.7 | 28.9 | 42.0 | 1282.7 | 173.8 | 3.4 | 5.8 | 108.4 | 118.6 |
| CC 13 | 1.0 | 5.8 | 58.9 | 32.5 | 39.6 | 1437.1 | 179.8 | 3.0 | 5.2 | 126.4 | 133.2 |
| CC 16 | 1.3 | 6.8 | 64.6 | 27.9 | 45.3 | 1406.1 | 179.9 | 3.0 | 5.3 | 120.8 | 128.5 |
| CC 18 | 1.6 | 5.3 | 58.6 | 28.4 | 42.5 | 1180.0 | 161.2 | 3.3 | 5.7 | 110.1 | 117.5 |
| CC 21 | 1.8 | 5.7 | 59.5 | 35.5 | 41.2 | 1378.1 | 168.4 | 5.0 | 4.1 | 112.2 | 119.0 |
| CC 22 | 2.0 | 4.3 | 59.5 | 31.0 | 39.8 | 1277.2 | 175.8 | 1.0 | 4.3 | 116.0 | 124.0 |
| CC 23 | 1.4 | 6.0 | 58.5 | 27.9 | 41.4 | 1116.1 | 158.9 | 2.5 | 4.8 | 123.8 | 131.4 |
| CC 25 | 2.0 | 6.0 | 55.4 | 28.3 | 38.5 | 1071.7 | 155.5 | 4.0 | 4.7 | 119.5 | 126.3 |
| CC 26 | 1.6 | 5.6 | 61.4 | 30.3 | 44.0 | 1301.6 | 169.5 | 1.3 | 6.1 | 114.1 | 121.3 |

**Table A2.** *Cont.*

| Accession | LflBroad | LflDent | StpLeng | StpWidt | StpLenT | StpArea | StpPerim | StpFlDen | PtlLenl | Flowr1 | FlwrFl |
|-----------|----------|---------|---------|---------|---------|---------|----------|----------|---------|--------|--------|
| CC 28 | 1.4 | 6.7 | 64.3 | 33.1 | 44.9 | 1449.4 | 176.0 | 2.7 | 5.5 | 117.3 | 124.1 |
| CC 29 | 1.9 | 4.4 | 57.1 | 27.6 | 40.0 | 1054.1 | 152.6 | 4.7 | 5.1 | 123.1 | 131.7 |
| CC 30 | 1.3 | 9.0 | 57.8 | 30.4 | 37.7 | 1202.6 | 169.7 | 2.7 | 4.6 | 115.5 | 122.7 |
| CC 31 | 1.5 | 7.0 | 54.8 | 27.8 | 38.5 | 1101.7 | 152.4 | 3.8 | 5.2 | 117.1 | 123.6 |
| CC 33 | 1.2 | 1.7 | 55.5 | 30.1 | 40.3 | 1180.3 | 158.1 | 4.3 | 5.4 | 109.7 | 114.2 |
| CC 35 | 1.3 | 6.8 | 48.6 | 25.6 | 33.9 | 884.6 | 134.4 | 4.0 | 4.8 | 114.1 | 119.0 |
| CC 36 | 1.6 | 2.3 | 57.6 | 27.8 | 40.1 | 1110.4 | 156.7 | 3.8 | 4.7 | 114.9 | 119.1 |
| CC 37 | 1.6 | 4.0 | 60.3 | 29.8 | 42.8 | 1263.7 | 169.3 | 4.8 | 5.4 | 110.4 | 119.4 |
| CC 38 | 1.4 | 4.0 | 59.4 | 32.3 | 39.9 | 1305.1 | 167.8 | 2.3 | 4.8 | 119.0 | 127.3 |
| CC 41 | 2.1 | 6.5 | 59.2 | 30.4 | 43.2 | 1294.6 | 179.2 | 3.0 | 6.1 | 123.9 | 130.5 |
| CC 42 | 1.6 | 3.3 | 61.7 | 32.4 | 43.2 | 1375.9 | 176.0 | 4.1 | 5.4 | 121.0 | 128.7 |
| CC 43 | 1.8 | 3.8 | 55.9 | 25.8 | 39.4 | 1001.0 | 150.8 | 3.8 | 4.8 | 117.8 | 124.0 |
| CC 45 | 1.9 | 5.3 | 60.4 | 28.3 | 43.3 | 1225.8 | 164.6 | 3.0 | 4.5 | 116.9 | 124.1 |
| CC 47 | 2.5 | 4.0 | 58.9 | 28.3 | 42.9 | 1182.2 | 156.9 | 4.8 | 4.9 | 122.4 | 129.5 |
| CC 48 | 2.1 | 6.5 | 52.6 | 26.8 | 37.2 | 958.0 | 148.3 | 3.3 | 4.6 | 111.0 | 120.6 |
| CC 49 | 2.3 | 3.7 | 53.9 | 27.8 | 39.1 | 1057.4 | 150.0 | 3.3 | 5.2 | 116.0 | 120.8 |
| CC 51 | 2.6 | 6.5 | 55.9 | 29.8 | 40.1 | 1155.7 | 155.6 | 3.3 | 5.3 | 110.6 | 117.3 |
| CC 52 | 2.4 | 6.5 | 65.8 | 33.8 | 45.3 | 1538.7 | 185.6 | 4.0 | 5.5 | 121.4 | 128.0 |
| CC 53 | 1.9 | 7.0 | 56.8 | 25.1 | 42.2 | 986.7 | 148.7 | 2.5 | 4.5 | 115.3 | 120.6 |
| CC 54 | 1.8 | 4.8 | 62.9 | 28.3 | 45.5 | 1305.0 | 170.4 | 3.0 | 5.1 | 106.8 | 116.0 |
| CC 55 | 2.4 | 5.0 | 62.7 | 34.5 | 46.1 | 1479.5 | 173.7 | 2.0 | 5.1 | 136.4 | 139.0 |
| CC 56 | 2.1 | 3.8 | 49.4 | 28.4 | 34.9 | 960.3 | 146.2 | 4.0 | 5.0 | 117.9 | 123.3 |
| CC 58 | 2.3 | 2.8 | 61.9 | 32.9 | 42.1 | 1353.1 | 169.9 | 3.0 | 4.0 | 93.0 | 98.5 |
| CC 59 | 2.0 | 5.0 | 47.5 | 25.8 | 33.6 | 802.3 | 130.4 | 4.7 | 4.2 | 109.7 | 116.4 |
| CC 62 | 1.9 | 5.8 | 54.8 | 28.0 | 39.0 | 1027.8 | 148.5 | 4.5 | 5.4 | 110.9 | 117.1 |
| CC 64 | 2.5 | 4.3 | 55.2 | 28.2 | 37.0 | 1087.9 | 151.8 | 3.3 | 3.9 | 128.2 | 132.5 |
| CC 67 | 1.9 | 6.8 | 64.3 | 34.9 | 43.3 | 1663.9 | 192.0 | 4.5 | 5.3 | 122.8 | 127.4 |
| CC 79 | 2.1 | 6.3 | 59.1 | 29.6 | 39.6 | 1264.8 | 163.4 | 5.3 | 4.6 | 125.9 | 130.8 |
| CC_ori | 1.8 | 5.4 | 51.4 | 26.4 | 34.9 | 938.3 | 142.6 | 4.4 | 4.6 | 125.1 | 130.8 |
| CA | 1.8 | 6.5 | 58.3 | 30.1 | 40.4 | 1155.5 | 162.3 | 3.3 | 4.0 | 124.3 | 131.6 |
| CE | 2.1 | 6.8 | 57.5 | 29.0 | 40.8 | 1153.1 | 160.7 | 3.5 | 4.7 | 131.4 | 136.5 |
| BM | n/r | n/r | 58.6 | 30.3 | 44.2 | 1231.6 | 164.9 | 3.3 | n/r | 107.9 | 114.0 |
| PS | 2.8 | 2.8 | 77.0 | 44.1 | 57.1 | 2720.2 | 228.0 | 4.8 | 5.6 | 99.5 | 104.8 |
| LSD | | | 7.38 | 4.91 | 5.38 | 348.5 | 23.71 | | 0.84 | 8.16 | 8.03 |
| *P<* | 0.001 | 0.001 | | | | | | 0.001 | | | |

**Table A2.** *Cont.*

| Accession | PlFlwNd | FlwWnC | FlwStC | FlwSeSh | PedSpur | PedSpLe | PedLeng1 | PedLeng2 | PodLeng | PodWidt | PodWall | PodShap |
|---|---|---|---|---|---|---|---|---|---|---|---|---|
| **CC 04** | 2.4 | 4.0 | 3.0 | 2.0 | 9.0 | 7.6 | 39.4 | 22.7 | 55.3 | 9.4 | 1.0 | 2.0 |
| **CC 05** | 2.1 | 4.0 | 3.0 | 1.9 | 9.0 | 12.6 | 62.8 | 17.5 | 49.1 | 9.1 | 1.0 | 1.9 |
| **CC 06** | 1.9 | 4.1 | 3.4 | 2.0 | 9.0 | 8.4 | 76.9 | 19.8 | 46.5 | 8.9 | 1.0 | 2.0 |
| **CC 07** | 2.1 | 4.0 | 3.0 | 1.9 | 9.0 | 8.6 | 72.3 | 19.2 | 44.6 | 8.9 | 1.0 | 2.0 |
| **CC 08** | 2.0 | 4.0 | 3.3 | 1.5 | 9.0 | 5.4 | 77.7 | 14.1 | 47.7 | 9.2 | 1.0 | 1.8 |
| **CC 10** | 2.5 | 4.1 | 3.5 | 1.8 | 9.0 | 5.0 | 57.3 | 19.3 | 48.9 | 9.4 | 1.0 | 1.4 |
| **CC 11** | 2.0 | 4.0 | 3.0 | 2.0 | 9.0 | 5.0 | 72.6 | 19.0 | 50.1 | 9.4 | 1.0 | 2.0 |
| **CC 13** | 2.4 | 4.2 | 2.8 | 1.6 | 9.0 | 6.4 | 76.1 | 23.6 | 47.1 | 8.6 | 1.0 | 2.0 |
| **CC 16** | 2.4 | 4.0 | 3.0 | 2.0 | 9.0 | 6.0 | 72.3 | 17.7 | 55.6 | 10.1 | 1.0 | 2.0 |
| **CC 18** | 2.0 | 4.0 | 3.0 | 2.0 | 9.0 | 3.1 | 64.0 | 17.8 | 49.6 | 8.9 | 1.0 | 1.9 |
| **CC 21** | 2.0 | 4.0 | 3.0 | 2.0 | 9.0 | 6.5 | 65.7 | 20.1 | 43.0 | 8.8 | 1.0 | 2.0 |
| **CC 22** | 2.0 | 4.0 | 3.0 | 1.8 | 9.0 | 3.2 | 68.6 | 24.4 | 45.3 | 8.7 | 1.0 | 2.0 |
| **CC 23** | 2.3 | 4.6 | 3.8 | 1.8 | 9.0 | 4.7 | 77.1 | 19.4 | 46.1 | 7.9 | 1.0 | 2.0 |
| **CC 25** | 2.1 | 4.0 | 3.0 | 2.0 | 9.0 | 5.1 | 50.6 | 12.9 | 48.8 | 8.8 | 1.0 | 2.0 |
| **CC 26** | 2.0 | 4.0 | 3.3 | 2.0 | 9.0 | 4.4 | 78.2 | 20.7 | 53.3 | 9.4 | 1.0 | 2.0 |
| **CC 28** | 2.3 | 4.0 | 3.0 | 1.7 | 9.0 | 6.3 | 71.5 | 20.5 | 46.9 | 8.6 | 1.0 | 1.9 |
| **CC 29** | 2.9 | 4.1 | 3.1 | 2.0 | 9.0 | 5.1 | 73.5 | 15.3 | 48.1 | 8.6 | 1.0 | 2.0 |
| **CC 30** | 2.0 | 4.0 | 3.0 | 2.0 | 9.0 | 4.4 | 48.5 | 17.3 | 48.3 | 8.8 | 1.0 | 1.8 |
| **CC 31** | 2.1 | 4.0 | 3.0 | 2.0 | 9.0 | 4.6 | 53.8 | 19.3 | 43.8 | 8.1 | 1.1 | 2.0 |
| **CC 33** | 2.3 | 4.0 | 3.0 | 2.0 | 7.4 | 3.0 | 41.1 | 13.0 | 38.1 | 9.3 | 1.0 | 2.0 |
| **CC 35** | 2.1 | 4.0 | 3.0 | 2.0 | 9.0 | 4.8 | 46.3 | 25.1 | 54.1 | 9.0 | 1.0 | 1.9 |
| **CC 36** | 2.6 | 3.9 | 3.0 | 2.0 | 9.0 | 7.1 | 59.6 | 21.4 | 47.3 | 9.3 | 1.0 | 1.9 |
| **CC 37** | 2.4 | 4.0 | 3.0 | 2.0 | 9.0 | 6.7 | 75.1 | 18.8 | 50.8 | 8.5 | 1.0 | 2.0 |
| **CC 38** | 2.6 | 4.0 | 3.0 | 2.0 | 9.0 | 15.8 | 66.6 | 22.9 | 45.1 | 9.6 | 1.0 | 2.0 |
| **CC 41** | 2.1 | 4.0 | 3.0 | 2.0 | 9.0 | 10.2 | 58.0 | 18.4 | 48.5 | 9.3 | 1.0 | 2.0 |
| **CC 42** | 2.9 | 4.0 | 3.0 | 2.0 | 9.0 | 14.4 | 85.8 | 28.2 | 49.0 | 9.2 | 1.0 | 2.0 |
| **CC 43** | 2.9 | 4.0 | 3.3 | 2.0 | 9.0 | 6.4 | 51.9 | 15.7 | 51.0 | 9.2 | 1.0 | 2.0 |
| **CC 45** | 2.3 | 4.0 | 3.0 | 2.0 | 9.0 | 7.5 | 48.3 | 20.3 | 48.3 | 9.4 | 1.0 | 2.0 |
| **CC 47** | 2.5 | 4.0 | 3.0 | 1.9 | 9.0 | 13.8 | 48.3 | 16.1 | 46.8 | 8.8 | 1.0 | 2.0 |
| **CC 48** | 2.1 | 4.0 | 3.5 | 2.0 | 9.0 | 5.1 | 46.1 | 13.3 | 50.6 | 9.4 | 1.0 | 2.0 |
| **CC 49** | 2.5 | 4.0 | 3.0 | 2.0 | 9.0 | 7.1 | 50.4 | 17.8 | 52.3 | 9.8 | 1.0 | 2.0 |
| **CC 51** | 2.6 | 4.0 | 3.0 | 2.0 | 9.0 | 6.3 | 48.3 | 15.8 | 54.9 | 9.3 | 1.0 | 2.0 |
| **CC 52** | 2.1 | 4.0 | 3.0 | 2.0 | 9.0 | 4.3 | 66.9 | 20.3 | 50.4 | 9.1 | 1.0 | 1.9 |
| **CC 53** | 2.8 | 4.0 | 3.3 | 1.9 | 9.0 | 5.6 | 45.7 | 16.9 | 57.4 | 10.1 | 1.0 | 2.0 |
| **CC 54** | 2.0 | 4.0 | 3.0 | 2.0 | 9.0 | 3.9 | 62.7 | 25.9 | 47.5 | 10.5 | 1.0 | 2.0 |

**Table A2.** *Cont*.

| Accession | PlFlwNd | FlwWnC | FlwStC | FlwSeSh | PedSpur | PedSpLe | PedLeng1 | PedLeng2 | PodLeng | PodWidt | PodWall | PodShap |
|-----------|---------|--------|--------|---------|---------|---------|----------|----------|---------|---------|---------|---------|
| CC 55 | 2.8 | 3.9 | 3.0 | 2.0 | 4.0 | 3.2 | 25.8 | 12.1 | 45.5 | 9.0 | 1.0 | 2.0 |
| CC 56 | 2.3 | 3.9 | 3.3 | 1.9 | 9.0 | 7.3 | 44.7 | 17.4 | 51.3 | 9.6 | 1.0 | 2.0 |
| CC 58 | 1.9 | 4.0 | 3.3 | 1.8 | 8.0 | 2.4 | 54.4 | 23.4 | 47.9 | 8.3 | 1.0 | 2.0 |
| CC 59 | 2.1 | 4.0 | 5.3 | 1.6 | 9.0 | 4.9 | 57.6 | 14.6 | 43.9 | 9.1 | 1.0 | 2.0 |
| CC 62 | 2.6 | 4.0 | 3.3 | 2.0 | 9.0 | 4.5 | 79.9 | 19.1 | 47.4 | 9.3 | 1.0 | 2.0 |
| CC 64 | 2.3 | 4.0 | 4.7 | 2.0 | 7.7 | 5.2 | 29.3 | 15.6 | 40.8 | 8.7 | 1.0 | 2.0 |
| CC 67 | 2.8 | 4.0 | 3.0 | 1.9 | 9.0 | 9.7 | 81.2 | 29.9 | 48.0 | 8.8 | 1.0 | 2.0 |
| CC 79 | 2.8 | 4.0 | 3.0 | 1.8 | 8.0 | 5.4 | 42.8 | 12.9 | 50.3 | 9.8 | 1.0 | 2.0 |
| CC_ori | 2.3 | 4.0 | 3.4 | 1.7 | 8.6 | 4.4 | 55.7 | 18.2 | 48.8 | 9.6 | 1.0 | 1.9 |
| CA | 2.3 | 4.0 | 3.3 | 1.8 | 8.0 | 5.2 | 40.3 | 14.4 | 47.6 | 9.4 | 1.0 | 2.0 |
| CE | 2.5 | 4.0 | 3.3 | 1.9 | 9.0 | 2.9 | 46.5 | 21.5 | 44.8 | 9.4 | 1.0 | 2.0 |
| BM | 2.0 | 2.0 | 1.0 | 2.0 | 6.0 | 4.9 | 63.7 | 17.6 | 63.7 | 11.1 | 2.0 | 2.0 |
| PS | 1.5 | 1.0 | 1.0 | 2.6 | 9.0 | 4.9 | 65.8 | 22.5 | 91.5 | 16.0 | 1.9 | 1.4 |
| LSD | 0.45 | | | 0.35 | | | 4.45 | 19.25 | 8.18 | 7.41 | 1.00 | |
| P< | | 0.001 | 0.001 | | 0.001 | | | | | | 0.001 | 0.001 |

| Accession | PodCurv | PodColor | PodOvul | PodNum | PodEmpt | SeedIntC | SeedShp | SeedStrc | SeedWrk | SeedCoC | SeedMar | SeedHilC |
|-----------|---------|----------|---------|--------|---------|----------|---------|----------|---------|---------|---------|----------|
| CC 04 | 1.2 | 2.0 | 7.2 | 12.0 | 0.0 | 3.0 | 0.89 | 1.0 | 1.8 | 3.0 | 5.0 | 5.0 |
| CC 05 | 2.0 | 2.0 | 7.5 | 11.1 | 0.0 | 2.8 | 0.89 | 1.0 | 1.3 | 3.0 | 4.5 | 5.0 |
| CC 06 | 3.0 | 2.0 | 7.4 | 9.3 | 0.0 | 3.0 | 0.88 | 1.0 | 2.5 | 3.0 | 4.8 | 7.5 |
| CC 07 | 1.3 | 2.0 | 7.4 | 20.4 | 0.0 | 2.7 | 0.90 | 1.0 | 1.3 | 3.0 | 5.0 | 4.4 |
| CC 08 | 1.3 | 2.0 | 6.5 | 12.7 | 0.0 | 3.0 | 0.90 | 1.0 | 1.3 | 3.0 | 5.0 | 9.0 |
| CC 10 | 2.0 | 2.0 | 7.4 | 13.6 | 0.1 | 2.8 | 0.89 | 1.0 | 3.0 | 3.0 | 5.0 | 4.0 |
| CC 11 | 1.8 | 2.0 | 7.6 | 17.6 | 1.2 | 3.0 | 0.89 | 1.0 | 3.0 | 3.0 | 5.0 | 5.0 |
| CC 13 | 1.8 | 2.0 | 7.8 | 12.2 | 0.0 | 3.0 | 0.90 | 1.0 | 1.8 | 3.0 | 5.0 | 2.6 |
| CC 16 | 1.8 | 2.0 | 7.4 | 11.3 | 0.3 | 1.8 | 0.90 | 1.0 | 3.5 | 3.0 | 1.5 | 6.0 |
| CC 18 | 2.0 | 2.0 | 7.5 | 12.6 | 1.3 | 2.8 | 0.90 | 1.0 | 2.0 | 3.0 | 1.0 | 5.0 |
| CC 21 | 1.0 | 2.0 | 7.2 | 13.8 | 0.2 | 2.3 | 0.90 | 1.0 | 1.0 | 3.0 | 5.0 | 3.0 |
| CC 22 | 1.7 | 2.0 | 6.8 | 12.3 | 0.0 | 2.7 | 0.89 | 1.0 | 1.3 | 3.0 | 3.7 | 3.7 |
| CC 23 | 4.0 | 2.0 | 6.9 | 12.4 | 0.3 | 2.0 | 0.89 | 1.0 | 1.3 | 2.8 | 4.5 | 9.0 |
| CC 25 | 1.8 | 2.0 | 7.6 | 13.5 | 0.4 | 2.3 | 0.89 | 1.0 | 1.8 | 3.0 | 4.8 | 3.5 |
| CC 26 | 2.4 | 2.0 | 7.9 | 13.7 | 0.7 | 2.7 | 0.89 | 1.0 | 2.4 | 3.0 | 4.7 | 7.9 |
| CC 28 | 2.1 | 2.0 | 6.7 | 11.7 | 0.1 | 2.4 | 0.90 | 1.0 | 1.6 | 3.0 | 5.0 | 2.7 |
| CC 29 | 2.4 | 2.0 | 7.9 | 10.3 | 0.0 | 2.4 | 0.89 | 1.0 | 2.4 | 3.0 | 1.0 | 2.7 |

**Table A2.** *Cont.*

| Accession | PodCurv | PodColor | PodOvul | PodNum | PodEmpt | SeedIntC | SeedShp | SeedStrc | SeedWrk | SeedCoC | SeedMar | SeedHilC |
|---|---|---|---|---|---|---|---|---|---|---|---|---|
| CC 30 | 3.0 | 2.0 | 8.0 | 9.5 | 0.0 | 3.0 | 0.90 | 1.0 | 1.3 | 3.0 | 5.0 | 5.0 |
| CC 31 | 2.6 | 2.9 | 6.0 | 15.1 | 0.0 | 1.6 | 0.89 | 1.0 | 2.8 | 3.0 | 5.0 | 5.5 |
| CC 33 | 1.4 | 2.0 | 7.2 | 4.8 | 0.8 | 4.0 | 0.90 | 1.0 | 1.8 | 2.5 | 5.0 | 5.0 |
| CC 35 | 2.0 | 2.0 | 7.9 | 14.3 | 2.8 | 3.0 | 0.89 | 1.0 | 3.0 | 3.0 | 5.0 | 4.0 |
| CC 36 | 2.3 | 2.0 | 7.0 | 9.6 | 0.5 | 2.0 | 0.89 | 1.0 | 1.5 | 3.0 | 5.0 | 3.0 |
| CC 37 | 3.0 | 2.0 | 8.1 | 17.5 | 1.6 | 3.0 | 0.89 | 1.0 | 2.3 | 3.0 | 4.8 | 2.5 |
| CC 38 | 2.1 | 2.0 | 7.7 | 16.1 | 1.6 | 3.0 | 0.89 | 1.0 | 1.9 | 3.0 | 5.0 | 9.0 |
| CC 41 | 1.0 | 2.0 | 7.8 | 12.5 | 0.9 | 1.5 | 0.89 | 1.0 | 2.5 | 3.0 | 5.0 | 3.5 |
| CC 42 | 3.9 | 2.0 | 8.0 | 10.7 | 0.3 | 1.0 | 0.89 | 1.0 | 1.9 | 3.0 | 5.0 | 8.4 |
| CC 43 | 2.3 | 2.0 | 8.4 | 14.8 | 1.0 | 1.0 | 0.89 | 1.0 | 2.0 | 3.0 | 5.0 | 1.5 |
| CC 45 | 2.0 | 2.0 | 7.4 | 14.1 | 0.4 | 1.3 | 0.89 | 1.0 | 1.8 | 3.0 | 4.0 | 4.5 |
| CC 47 | 3.0 | 2.0 | 7.0 | 13.3 | 1.8 | 2.4 | 0.89 | 1.0 | 2.3 | 3.0 | 3.9 | 5.6 |
| CC 48 | 2.0 | 2.0 | 7.6 | 16.5 | 1.1 | 1.6 | 0.90 | 1.0 | 1.0 | 3.0 | 3.5 | 4.5 |
| CC 49 | 2.0 | 2.0 | 7.8 | 21.8 | 2.0 | 3.0 | 0.90 | 1.0 | 3.0 | 3.0 | 3.3 | 4.3 |
| CC 51 | 1.8 | 2.0 | 8.4 | 16.0 | 1.8 | 3.0 | 0.90 | 1.0 | 1.5 | 3.0 | 5.0 | 3.5 |
| CC 52 | 1.3 | 2.0 | 8.1 | 16.1 | 0.8 | 3.0 | 0.89 | 1.0 | 3.0 | 3.0 | 3.0 | 6.0 |
| CC 53 | 4.0 | 2.0 | 8.8 | 14.5 | 3.9 | 3.0 | 0.90 | 1.0 | 1.5 | 3.0 | 5.0 | 9.0 |
| CC 54 | 2.0 | 2.0 | 7.8 | 22.1 | 0.1 | 3.0 | 0.90 | 1.0 | 2.3 | 3.0 | 4.8 | 5.5 |
| CC 55 | 1.8 | 2.0 | 7.2 | 9.4 | 0.2 | 1.4 | 0.90 | 1.0 | 3.0 | 3.0 | 1.0 | 8.2 |
| CC 56 | 1.5 | 2.0 | 7.0 | 13.1 | 2.5 | 1.9 | 0.90 | 1.0 | 2.0 | 3.0 | 1.0 | 5.0 |
| CC 58 | 1.5 | 2.0 | 6.9 | 9.5 | 0.0 | 2.2 | 0.89 | 1.0 | 1.3 | 3.0 | 5.0 | 4.0 |
| CC 59 | 2.7 | 2.0 | 7.4 | 15.3 | 0.6 | 1.8 | 0.90 | 1.0 | 1.3 | 3.0 | 5.0 | 4.4 |
| CC 62 | 3.3 | 2.0 | 7.0 | 17.8 | 1.3 | 1.9 | 0.89 | 1.0 | 1.8 | 3.0 | 4.8 | 5.0 |
| CC 64 | 2.6 | 2.0 | 6.3 | 11.7 | 0.5 | 2.7 | 0.90 | 1.0 | 3.0 | 3.0 | 1.0 | 5.0 |
| CC 67 | 2.5 | 2.0 | 6.9 | 11.9 | 0.0 | 4.8 | 0.90 | 1.0 | 2.3 | 2.8 | 5.0 | 4.0 |
| CC 79 | 1.6 | 1.9 | 7.1 | 13.4 | 0.7 | 2.0 | 0.89 | 1.0 | 2.4 | 3.0 | 5.0 | 3.3 |
| CC_ori | 1.9 | 2.0 | 7.4 | 10.4 | 1.0 | 2.7 | 0.89 | 1.0 | 2.2 | 3.0 | 4.3 | 4.1 |
| CA | 2.3 | 2.0 | 7.1 | 13.3 | 0.6 | 2.8 | 0.89 | 1.0 | 3.3 | 3.0 | 4.0 | 5.0 |
| CE | 2.0 | 2.0 | 7.2 | 9.5 | 0.8 | 1.5 | 0.88 | 1.0 | 3.0 | 3.0 | 4.3 | 5.7 |
| BM | 1.8 | 2.0 | 6.8 | 6.6 | 0.4 | 4.5 | 0.86 | 1.0 | 2.3 | 1.1 | 1.0 | 5.5 |
| PS | 1.8 | 2.1 | 8.4 | 5.6 | 0.0 | 2.8 | 0.85 | 1.9 | 8.5 | 1.3 | 1.5 | 4.0 |
| LSD | | | 0.77 | 4.67 | 1.07 | | 0.008 | | | | | |
| P< | 0.001 | 0.001 | | | | 0.001 | | 0.001 | 0.001 | 0.001 | 0.001 | 0.001 |

**Table A2.** *Cont.*

| Accession | SeedNPla | SeedYld | HSW | SeedLen | SeedWid | SeedArea | SeedPeri | SeedLW |
|---|---|---|---|---|---|---|---|---|
| CC 04 | 57.2 | 4.7 | 10.5 | 5.7 | 5.1 | 22.1 | 17.6 | 1.11 |
| CC 05 | 44.1 | 4.3 | 11.0 | 5.8 | 5.2 | 23.3 | 18.0 | 1.11 |
| CC 06 | 25.4 | 2.9 | 11.0 | 5.5 | 4.9 | 20.4 | 16.9 | 1.12 |
| CC 07 | 77.4 | 4.5 | 9.0 | 5.2 | 4.7 | 18.8 | 16.2 | 1.12 |
| CC 08 | 46.7 | 3.9 | 8.9 | 5.2 | 4.8 | 19.1 | 16.3 | 1.08 |
| CC 10 | 43.4 | 3.5 | 8.8 | 5.4 | 4.8 | 19.7 | 16.6 | 1.11 |
| CC 11 | 64.0 | 4.8 | 9.6 | 5.5 | 5.0 | 20.6 | 17.0 | 1.11 |
| CC 13 | 51.2 | 3.4 | 8.0 | 5.0 | 4.5 | 17.4 | 15.5 | 1.10 |
| CC 16 | 39.8 | 4.0 | 10.5 | 5.7 | 5.1 | 22.0 | 17.5 | 1.12 |
| CC 18 | 46.0 | 4.4 | 10.5 | 5.6 | 5.1 | 21.5 | 17.3 | 1.10 |
| CC 21 | 45.2 | 3.8 | 10.0 | 5.3 | 4.9 | 19.8 | 16.6 | 1.09 |
| CC 22 | 43.8 | 4.9 | 11.8 | 5.9 | 5.4 | 23.6 | 18.3 | 1.09 |
| CC 23 | 40.0 | 3.1 | 8.4 | 5.2 | 4.7 | 18.4 | 16.0 | 1.11 |
| CC 25 | 48.0 | 3.9 | 9.4 | 5.3 | 4.8 | 19.0 | 16.3 | 1.10 |
| CC 26 | 48.3 | 5.0 | 11.2 | 5.8 | 5.3 | 23.3 | 18.1 | 1.09 |
| CC 28 | 41.3 | 3.5 | 8.1 | 5.0 | 4.6 | 17.8 | 15.7 | 1.10 |
| CC 29 | 31.1 | 3.0 | 9.8 | 5.4 | 5.0 | 20.5 | 16.9 | 1.09 |
| CC 30 | 42.8 | 4.3 | 10.3 | 5.6 | 5.1 | 21.6 | 17.4 | 1.08 |
| CC 31 | 47.9 | 4.8 | 10.9 | 5.5 | 5.1 | 21.1 | 17.1 | 1.10 |
| CC 33 | 16.4 | 1.7 | 7.2 | 4.7 | 4.1 | 15.1 | 14.2 | 1.15 |
| CC 35 | 61.0 | 4.4 | 9.0 | 5.3 | 4.8 | 19.8 | 16.6 | 1.11 |
| CC 36 | 38.3 | 3.2 | 9.3 | 5.4 | 5.0 | 20.0 | 16.7 | 1.08 |
| CC 37 | 65.8 | 4.9 | 10.1 | 5.5 | 5.0 | 20.9 | 17.1 | 1.11 |
| CC 38 | 56.3 | 4.0 | 9.4 | 5.4 | 4.9 | 20.0 | 16.7 | 1.09 |
| CC 41 | 49.8 | 3.8 | 9.7 | 5.6 | 4.9 | 21.0 | 17.1 | 1.14 |
| CC 42 | 40.0 | 4.8 | 12.2 | 6.0 | 5.4 | 24.8 | 18.6 | 1.11 |
| CC 43 | 63.3 | 4.9 | 9.9 | 5.6 | 5.0 | 21.5 | 17.4 | 1.12 |
| CC 45 | 57.8 | 4.7 | 10.0 | 5.6 | 5.1 | 21.7 | 17.4 | 1.12 |
| CC 47 | 44.5 | 4.1 | 9.1 | 5.4 | 4.9 | 19.6 | 16.5 | 1.11 |
| CC 48 | 62.4 | 4.7 | 10.2 | 5.6 | 5.0 | 21.3 | 17.2 | 1.11 |
| CC 49 | 65.7 | 5.7 | 11.8 | 5.8 | 5.4 | 23.9 | 18.2 | 1.09 |
| CC 51 | 76.4 | 5.6 | 11.2 | 5.7 | 5.2 | 22.4 | 17.7 | 1.09 |
| CC 52 | 63.3 | 4.8 | 9.8 | 5.6 | 5.0 | 21.0 | 17.1 | 1.12 |
| CC 53 | 65.9 | 5.1 | 10.5 | 5.6 | 5.1 | 21.7 | 17.4 | 1.11 |
| CC 54 | 87.1 | 4.2 | 8.5 | 5.3 | 4.8 | 19.5 | 16.4 | 1.12 |

**Table A2.** *Cont.*

| Accession | SeedNPla | SeedYld | HSW | SeedLen | SeedWid | SeedArea | SeedPeri | SeedLW |
|-----------|----------|---------|------|---------|---------|----------|----------|--------|
| **CC 55** | 32.2 | 3.3 | 9.6 | 5.4 | 5.0 | 20.6 | 16.9 | 1.10 |
| **CC 56** | 57.1 | 5.1 | 10.4 | 5.7 | 5.1 | 22.0 | 17.5 | 1.12 |
| **CC 58** | 43.5 | 4.1 | 11.3 | 5.6 | 5.1 | 21.5 | 17.4 | 1.10 |
| **CC 59** | 53.3 | 4.7 | 10.0 | 5.4 | 5.0 | 20.4 | 16.8 | 1.09 |
| **CC 62** | 55.5 | 5.6 | 11.5 | 5.7 | 5.3 | 22.9 | 17.9 | 1.09 |
| **CC 64** | 41.0 | 4.0 | 9.4 | 5.3 | 4.8 | 19.3 | 16.3 | 1.11 |
| **CC 67** | 43.8 | 3.9 | 9.1 | 5.4 | 4.9 | 20.1 | 16.8 | 1.11 |
| **CC 79** | 46.9 | 4.7 | 11.3 | 5.8 | 5.2 | 22.9 | 17.9 | 1.11 |
| **CC_ori** | 35.4 | 3.0 | 9.2 | 5.4 | 4.9 | 20.2 | 16.8 | 1.11 |
| **CA** | 46.5 | 4.0 | 9.2 | 5.5 | 5.0 | 20.8 | 17.0 | 1.11 |
| **CE** | 35.0 | 3.0 | 9.7 | 5.7 | 5.1 | 21.6 | 17.4 | 1.12 |
| **BM** | 21.8 | 3.3 | 21.8 | 7.0 | 6.5 | 33.6 | 22.0 | 1.08 |
| **PS** | 27.3 | 6.2 | 24.3 | 8.5 | 6.9 | 45.8 | 25.9 | 1.24 |
| **LSD** | 15.81 | 1.25 | 2.08 | 0.40 | 0.36 | 3.05 | 1.26 | 0.026 |
| *P<* | | | | | | | | |

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
