# Peer review of "The Genetic Structure of the Field Pea Landrace “Roveja di Civita di Cascia”"

_sustainability, doi:10.3390/su11226493_

Round 1

Reviewer 1 Report

sustainability-637089

After careful review of the manuscript entitled “The genetic structure of the field pea landrace Roveja di Civita di Cascia” – Ms ID: sustainability-637089, I am very pleased to recommend it for publication in Sustainability.

I believe that the results fall within the scope of the Journal and they will be very useful to improve the sustainable use of this landrace. This work fits with the increase of global demand for novel food rich in proteins in addition to the urgent requirement to reduce the impact of agricultural practices on the environment.

However, there are some minor issues that I would represent to the authors.

Below the point by point list. Moreover an English language editing should be done in the manuscript.

Introduction

L26: check plural for “divesifications”

L56: in my opinion the reference [16] is not appropriate

L63: check English, use “facilitates”

L70: describe acronym SSR

M&M

L76-77: precise the name of species under investigation in “original plant population”

L90: check the name of Table A1 at L366 pag. 12, in appendix A or B?

L93: use plural for “measurement”

R&D

L160 and L299: I would prefer a unique paragraph for “Results and discussion” section, because the results are described and discussed here.

L174: change “were” to “was”

L211 and L244: merge the two paragraph 3.2 and 3.3 or rename them

L224: check the name PeaCPLHPPS in the sequence of Figure 1S of pag. 11, in this page rename Figure 1 to Figure 1S.

L228: check the motif (GA) for locus AA321, it should be (TC)

L239: insert “alleles” after “17”

L252: check English, for “the reproductive system of the species is likely to be based on high rates”

L268: there are 253 individuals analyzed by SSR, it would be better to indicate the same value at L100-102 of M&M

L296: check English for “the landrace “Roveja di Civita di Cascia” is likely to be a blend”

After L298: in Tab. 4, if it is used the length of SSR, then modify the title “The values are number of repeats.”

L299: change this title to (for example) “Sustainable landrace conservation”

L327, 328: change “size” to “extent”

L333: check English for “and, eventually, assess their changes caused by natural evolutive factors”

L336: delete “low”

L337: delete “at most”

L343-344: the sentence is not clear, check English and explain better: “In some cases they are asked by public institutions to safeguard them against incentives provided as subsidies.”

L347: delete “However”

L349-350: delete this sentence: “Similar case can be described for the lentil of “Castelluccio di Norcia”, another village in Valnerina that has not yet achieved the same reward.”

L353: delete “of marginal lands”

Check the names of traits in Table A2: PtlLeng, PodLenl, IntColSe

Author Response

Introduction

L26: check plural for “divesifications”

Done

L56: in my opinion the reference [16] is not appropriate

The reference is appropriate because in the same document concerning saffron is reported a note on “Roveglia”. We have the original document and could provide them. Unfortunately, they are in italian, if necessary we can translate it

L63: check English, use “facilitates”

Done

L70: describe acronym SSR

Done in the abstract and in the key words

M&M

L76-77: precise the name of species under investigation in “original plant population”

Done

L90: check the name of Table A1 at L366 pag. 12, in appendix A or B?

Done

L93: use plural for “measurement”

Done

R&D

L160 and L299: I would prefer a unique paragraph for “Results and discussion” section, because the results are described and discussed here.

The template of Sustainability is based on two separate sections. Actually we did some discussion in the results in order to make easy its readibility, while in the following section we discuss with the focus of sustainability. We would like to maintain the paper organization unless the Editor suggests otherwise

L174: change “were” to “was”

Done

L211 and L244: merge the two paragraph 3.2 and 3.3 or rename them

Done and renamed accordingly

L224: check the name PeaCPLHPPS in the sequence of Figure 1S of pag. 11, in this page rename Figure 1 to Figure 1S.

Done

L228: check the motif (GA) for locus AA321, it should be (TC)

Done

L239: insert “alleles” after “17”

Done

L252: check English, for “the reproductive system of the species is likely to be based on high rates”

Done

L268: there are 253 individuals analyzed by SSR, it would be better to indicate the same value at L100-102 of M&M

Done at line 104

L296: check English for “the landrace “Roveja di Civita di Cascia” is likely to be a blend”

Done

After L298: in Tab. 4, if it is used the length of SSR, then modify the title “The values are number of repeats.”

Done

L299: change this title to (for example) “Sustainable landrace conservation”

See comment above on Sustainability’s template

L327, 328: change “size” to “extent”

Done

L333: check English for “and, eventually, assess their changes caused by natural evolutive factors”

Done

L336: delete “low”

Done

L337: delete “at most”

Done

L343-344: the sentence is not clear, check English and explain better: “In some cases they are asked by public institutions to safeguard them against incentives provided as subsidies.”

Done

L347: delete “However”

Done

L349-350: delete this sentence: “Similar case can be described for the lentil of “Castelluccio di Norcia”, another village in Valnerina that has not yet achieved the same reward.”

Done

L353: delete “of marginal lands”

Done

Check the names of traits in Table A2: PtlLeng, PodLenl, IntColSe

Done

Reviewer 2 Report

This manuscript is a  description of the importance of the genetic material of landraces. The manuscript is well written and easy to read. The aims are well explained. The experiments are well designed and robust, as well as the statistical analyses.

The interpretations are supported by the results, and the Discussion  is not just a summery of the results.

There are only few minor suggestions. My suggestions are indicated in the accompanying document.

Author Response

L14: Add the name of the control entries

Done

L17: Change “Phenotipic” with “Phenotypic”

Done

L87: Add the type of soil used in pots

Done

L183: Change “former and latter” with “Roveja and Paladio”

Done

L201: Write all the name and not only the abbreviation. Change black color in Figure 1

The purpose of the figure was to represent the magnitude of the variability of the individual lines of Roveja in comparison with the controls. Control symbols are on a higher layer and therefore they overlap the Roveja symbols, making easier the graph interpretation. We tried to change colors but the result was definitely lower. We added full names, the same for Figure 2 in lines 277-279

L237: Explain how did you go from 15 to 5 SSRs

3 out of 15 SSR markers were discarded because the repetitive motif was not found. Of the remaining 12 we used only the most 5 polymorphic after a preliminary screening, still enough to assess the genetic structure of the landrace. We added in line 131 M&M the number of SSRs used

Reviewer 3 Report

The paper is focused on the determination of the genetic variation of one of the landraces of Pisum sativum. It has been shown that significant differences could be observed compared with the control ones.

The paper originality is somehow limited as such determination has been done (see for example Burstin et al. BMC Genomics 2015; 16(1): 105.) but at least the study is important for Italian landraces. The MS is clearly on the molecular biology subject but the authors' tried to discuss sustainability. In this regard, I suggest that to shorted the molecular part in the results and to increase the discussion about sustainability applied in agriculture.   

In order to the paper to become more interesting for readers, I suggest being included some data regarding yield and production.

Author Response

In order to the paper to become more interesting for readers, I suggest being included some data regarding yield and production.

We think that the molecular part is important for the traceability in case of frauds (see added sentence in lines 337-339) and to investigate the genetic structure of this landrace without environmental influences.

The aim of the paper was not the assessment of Roveja productivity, rather its sustainability in time. At the end of the discussion we added few lines concerning yields and economic return.